# A Brain-Inspired Machine Learning Paradigm for Nature-Powered Equation Solving

## Abstract

Solving equations is fundamental to human understanding of the world. While modern machine learning methods are powerful equation solvers, their escalating complexity and extreme operational costs hinder sustainable development. In contrast, nature effortlessly solves complex equations through dynamical systems that instinctively evolve to low-energy states without explicit instructions. However, existing attempts to leverage dynamical systems are limited by low expressivity and a lack of training support. To this end, we propose DS-Solver, a nature-powered AI paradigm employing an expressive, self-trainable dynamical system capable of accurately solving a wide spectrum of equations with extraordinary efficiency. (1) We enhance system expressivity by enriching node dynamics with coupled real-valued and polarized shadow nodes, capturing complex interactions inherent in the real world. (2) We propose an on-device learning method that leverages intrinsic electrical signals as loss, enabling the dynamical system to instantly train itself at negligible cost. Experimental results across key equations from diverse domains demonstrate that DS-Solver achieves 42% higher accuracy than current SOTA – while offering orders-of-magnitude improvements in speed and energy efficiency over traditional neural network solutions on GPUs for both inference and training, showcasing its broader impact in overcoming persistent computational bottlenecks across various critical fields.

## 1 Introduction

Solving equations is at the heart of human understanding, allowing us to describe society, the universe, and reality, and enabling us to anticipate future events. Modern ML methods, particularly neural networks, have played a critical role as powerful equation solvers. By observing data, these models approximate equations into data distributions represented by carefully designed neural networks, finding high-probability solutions from the learned distributions as solutions to the equations. However, the skyrocketing complexity of models programmed on general-purpose processors (e.g. GPU) with a tremendous number of explicit instructions has led to extreme operational costs, especially training, hindering the sustainable development of AI.

In contrast, nature effortlessly and constantly solves complex equations, as seen in dynamical systems. Consider partial differential equations (PDEs) in molecular dynamics and chemical reactions: dynamical systems solve them by representing the underlying data distributions as energy landscapes, where lower energy states indicate higher probability. Driven by their intrinsic nature (Second Law of Thermodynamics), dynamical systems instinctively evolve to the lowest energy state at equilibrium – a process called ***natural annealing*** – thus finding the solutions to the equations. Sharing a similar statistical basis with ML, this method is nature-powered and operates without explicit instructions, ensuring extreme efficiency. Notably, numerous high-profile scientific studies (Friston, 2010; Inagaki et al., 2019; Wills et al., 2005) reveal that the brain functions as a dynamical system, seamlessly integrating inference and training by continually settling into stable, low-energy states representing cognitive processes and memories. This insight partially explains the remarkable efficiency of biological intelligence. These observations raise a compelling question: ***Can dynamical systems serve as AI supercomputers, creating a nature-powered ML paradigm that solves equations with efficiency comparable to biological intelligence?*** The first challenge is to make dynamical systems controllable and programmable.

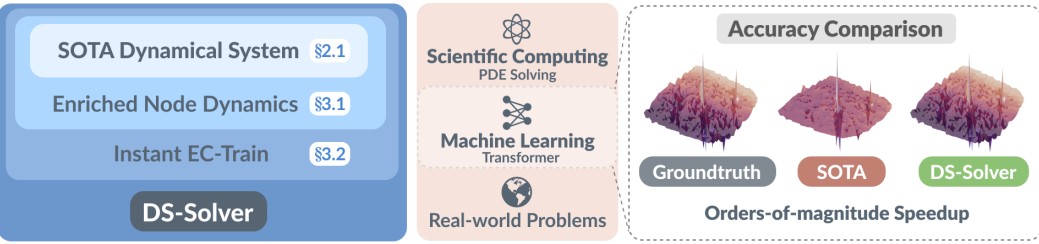

a. Overview of the proposed DS-Solver          b. Transformer Approximation

Figure 1: The overview of the proposed DS-Solver.

Recognizing their huge potential, researchers have developed dynamical systems that offer easy control and programmability (Böhm et al., 2022; Moy et al., 2022; Lo et al., 2023), primarily composed of electronic components like resistors and capacitors. In the past two years, these dynamical systems have been utilized to solve simple learning problems, such as traffic prediction (Pan et al., 2023; Wu et al., 2024) and collaborative filtering (Liu et al., 2023). Specifically, these systems are governed by parameterized Hamiltonians, which determine the complexity of the energy landscape and thus the system's expressivity. The programmability stems from adjustable resistors, whose conductance represents Hamiltonian parameters, collectively shaping the exact structure of the energy landscape. To solve learning problems, existing works (Pan et al., 2023; Liu et al., 2023; Wu et al., 2024) employ traditional machine learning training methods, e.g. stochastic gradient descent (SGD) on GPUs, to find the optimal parameters that align the constructed energy landscape with the data distribution. During inference, these parameters are programmed into the resistors, enabling the dynamical system to leverage natural annealing to find the energy minimum as the desired solution. These approaches enable dynamical systems as efficient equation solvers, capable of finding solutions through natural annealing with no explicit instructions involved in inference, thus approaching the efficiency of intelligence observed in nature.

Unfortunately, the applicability and broader impact of electronic dynamical systems are significantly limited due to two main challenges: *1. Low Expressivity:* Existing works employ dynamical systems governed by a quadratic Hamiltonian, leading to low-rugosity energy landscapes with only linear interactions among nodes, and hence limiting accuracy in real-world contexts. The current SOTA (Wu et al., 2024) can solve linear equations like matrix multiplication with satisfactory accuracy but fails when tackling more advanced problems such as PDEs (Table 1) and transformers (as shown in Figure 1.b) that dominate scientific computing and machine learning, respectively. *2. Lack of Support for Training:* Present approaches realize inference on dynamical systems through on-device natural annealing; however, the training process to construct the desired energy landscape must be performed entirely on digital processors. This results in even higher training costs than traditional DNNs due to the intrinsic complexity of dynamical systems, even with simple Hamiltonians. This decoupled training and inference depart from the intelligence observed in nature and prevent this new AI paradigm from addressing the most critical problem in AI development – extreme training cost. Directly utilizing dynamical systems to perform nature-powered training at ultra-low cost is crucial and insightful.

To address these bottlenecks and realize the potential of electronic dynamical systems, we introduce *DS-Solver* – a nature-powered AI paradigm that employs an expressive, self-trainable dynamical system to solve a wide spectrum of equations with superior accuracy and unprecedented efficiency. Specifically, we enhance the dynamical system through two key innovations: (1) We significantly improve system expressivity by enriching the node dynamics with tightly coupled real-valued and polarized shadow nodes, enabling the precise capture of high-order and highly nonlinear interactions. (2) We propose an on-device self-learning method that allows the dynamical system to leverage its intrinsic electrical signals as loss (akin to the brain), enabling it to self-construct its energy landscape, align with the target distribution, and achieve instant training at negligible costs. Unlike digital processors that orchestrate electrons following explicit instructions from AI programs, DS-Solver performs both inference and training by allowing electrons to instinctively seek equilibrium. This approach offers a unique, nature-powered AI paradigm with 'electron speed' and ultra-low power consumption, outperforming GPUs by orders of magnitude in both speed and energy efficiency, paving the way to the efficiency of biological intelligence. By significantly expanding the applicability of dynamical systems to encompass representative challenges from diverse domains,

e.g. PDEs in scientific computing, transformers in ML, and even hard-to-define equations in complex real-world problems like pandemic propagation, DS-Solver holds the potential to overcome persistent computational bottlenecks and drive advancements across various fields. The major contributions of this paper are summarized as follows:

- We propose DS-Solver, a nature-powered AI paradigm harnessing dynamical systems with collocated inference & training to accurately and efficiently solve key equations across diverse domains.
- We enhance the expressivity of existing dynamical-system AI paradigms by introducing enriched node dynamics coupled with polarized shadow nodes, enabling high-rugosity energy landscapes that precisely capture complex node interactions in the real world.
- We propose an on-device training method that enables dynamical systems to self-construct energy landscapes using internal electrical signals, allowing for second-level training at negligible cost.
- Experimental results demonstrate that DS-Solver solves equations with high accuracy, achieving orders of magnitude speedup ($\sim 10^3 \times$) and energy efficiency ($\sim 10^5 \times$) over A100 GPU.

## 2 BACKGROUND AND RELATED WORK

### 2.1 BACKGROUND

This section provides an overview of the current state-of-the-art (SOTA) dynamical system designs employed in solving learning problems (Wu et al., 2024). We begin with the dynamical system model, proceed to its precise hardware embodiment, and conclude with its training methods.

**Dynamical System Model.** A dynamical system is a mathematical model that describes how elements influence each other's states over time, causing the system to evolve, often toward equilibrium. These systems feature an energy landscape defined by a Hamiltonian, with energy minima at equilibrium states. The Hamiltonian of the current SOTA dynamical system used in AI is defined:

$$\mathcal{H}_{rv} = -\sum_{i \neq j}^{N} J_{ij}\sigma_i\sigma_j + \sum_{i=1}^{N} h_i\sigma_i^2, \quad \sigma_i, \sigma_j \in [-1, 1] \subset \mathbb{R}. \tag{1}$$

Here, $J_{ij}$ represents the interaction strength between two nodes $\sigma_i$ and $\sigma_j$, and $h_i$ denotes the self-reaction strength of $\sigma_i$ to external influences. Assuming a Boltzmann distribution $p_{rv} = e^{-\beta \mathcal{H}_{rv}}/Z$, where the partition function $Z$ serves as a normalizing constant ensuring that probabilities sum to one, the energy landscape is mapped to a probability distribution, with the lowest energy state corresponding to the highest probability state.

The spontaneous energy decrease of the system is guaranteed by the carefully designed node dynamics, which dictate how the system evolves over time. The current approach designs the node dynamics $d\sigma_i/dt$ as follows:

$$\frac{d\sigma_i}{dt} = -\frac{\partial \mathcal{H}_{rv}}{\partial \sigma_i} = \sum_{j \neq i}^{N} (J_{ij} + J_{ji})\sigma_j - 2h_i\sigma_i, \tag{2}$$

This node dynamics adheres to Lyapunov stability analysis, guaranteeing that the system evolves towards the lowest energy state:

$$\frac{d\mathcal{H}_{rv}}{dt} = \sum_{i=1}^{N} \left( \frac{\partial \mathcal{H}_{rv}}{\partial \sigma_i} \frac{d\sigma_i}{dt} \right) \leq 0. \tag{3}$$

**Dynamical System Embodiment as a Processor.** In the current SOTA, the dynamical system governed by the Hamiltonian defined in equation 1 is effectively embodied as a low-power processor composed of programmable electronic components, such as resistors and capacitors, as illustrated in Figure 2. The key idea behind this embodiment is to precisely and efficiently realize the node dynamics using electronic components, ensuring that the system's energy decreases spontaneously. In this design, each node $\sigma_i$ is implemented as a nanoscale capacitor within a node unit ($N_i$), with its voltage representing the node value. Each capacitor is coupled with a resistor of resistance $R_i$ set to $1/(2h_i)$, forming a resistive current within the node unit. This current acts as an energy regulator, realizing the term $2h_i\sigma_i$ from the node dynamics and enabling real-valued stability. Furthermore, each pair of capacitors from different node units ($N_i$ and $N_j$) is structurally connected by a programmable resistor in the coupling unit ($CU_{ij}$) with resistance $R_{ij}$ set to $1/J_{ij}$. This effectively incorporates the term $\sum_{j \neq i}^{N} (J_{ij} + J_{ji})\sigma_j$ from the node dynamics into a resistively coupled capacitor network.

**Training of Dynamical System.** The training process of a dynamical system is to find the optimal parameters $\mathbf{J}$ and $\mathbf{h}$ in the Hamiltonian $\mathcal{H}_{rv}$ to construct an energy landscape that mirrors the target data distribution. Prior works have trained the model using computationally expensive traditional statistical methods executed on digital processors, mainly GPUs. Specifically, the training process begins by estimating the lowest energy states of the dynamical system using methods such as conditional likelihood maximization (Wu et al., 2024) and the contrastive divergence algorithm (Hinton, 2002). The discrepancies between the estimations and the ground truths are evaluated using metrics such as Mean Absolute Error (MAE). These metrics serve as loss functions to update the model parameters, thereby reconstructing the energy landscape to align the ground truth with the system's energy minima. During inference, natural annealing drives the system toward the lowest energy state, allowing it to find the solution with the highest likelihood for the target problem.

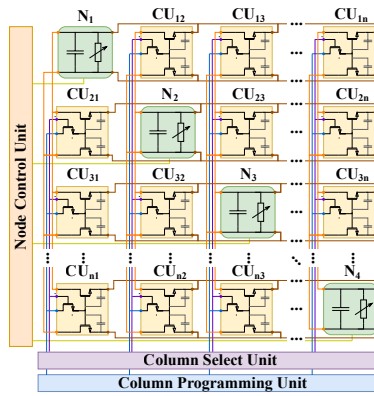

Figure 2: The overall architecture of the electronic dynamical system.

## 2.2 RELATED WORK

Dynamical systems as efficient supercomputers have gained significant attention in recent years, particularly for solving optimization problems. The Ising machine, one of the earliest processors to harness dynamical systems for such tasks, embodies the Ising model originally developed for ferromagnetism in statistical physics. Ising machines have demonstrated breakthrough efficiency in solving numerous NP-complete binary optimization problems, with results published in prominent scientific journals (Mohseni et al., 2022; Lo et al., 2023; Böhm et al., 2019). For instance, researchers have employed Ising machines to tackle satisfiability (SAT) problems (Sharma et al., 2023a;b), as well as MAX-CUT and graph coloring problems (Wang & Roychowdhury, 2019; Böhm et al., 2019). Recognizing their potential, scientists have also explored the use of dynamical systems for ML, addressing real-world issues such as traffic congestion (Pan et al., 2023), collaborative filtering (Liu et al., 2023), and neural network training (Böhm et al., 2022).

While these studies provide valuable insights into leveraging dynamical systems for ML tasks, their scope and applicability are limited by the binary nature of Ising machine nodes, hindering progress in more complex, real-valued scenarios. To address this binary limitation, Wu et al. (Wu et al., 2024) proposed an extension of the binary Ising model to accommodate real-valued nodes and developed a real-valued Ising machine for accelerated inference in graph learning problems, setting the current SOTA for dynamical system approaches in ML. However, their contributions are constrained by two key limitations. First, while their proposed Hamiltonian supports real-valued nodes, it only accounts for linear node interactions, which is insufficient for capturing the intrinsic nonlinearity present in many complex problems. Second, their approach utilizes the power of dynamical systems exclusively during the inference phase, leaving the computationally intensive training process unaddressed. These limitations, which significantly constrain the broader impact of dynamical systems in ML, will be addressed in this work.

## 3 METHODOLOGY

The proposed DS-Solver features two highlights: expressivity enhancement and instant on-device training, detailed in Section 3.1 and Section 3.2, respectively.

## 3.1 EXPRESSIVITY ENHANCEMENT

***The SOTA Dynamical System Model.*** Current SOTA dynamical system model is governed by its Hamiltonian, as illustrated in equation 1, with node dynamics described in equation 2. The system stabilizes when each node's dynamics converges to zero, and at this equilibrium, the lowest energy state reveals a linear interaction between nodes:

$$\sigma_i = \frac{1}{2h_i} \sum_{j \neq i}^{N} \left( J_{ij} + J_{ji} \right) \sigma_j \qquad (4)$$

While the system exhibits inherent nonlinearity due to the voltage range limitations of the capacitors, the linear inter-node interaction significantly constrains the dynamical system's capacity to address more complex equations.

***DS-Solver Model with Enriched Node Dynamics.*** Drawing inspiration from the Ising model – a binary dynamical system model renowned for its high expressivity and effectiveness in modeling complex physical systems – we propose to enhance the expressivity of dynamical systems by enriching node dynamics. Specifically, we selectively couple real-valued nodes with their corresponding polarized shadow nodes, augmenting the system with node dynamics powered by duo-channel (main channel from real-valued node and side-channel from polarized one) inter-node interactions. Each shadow node is strongly connected with its parent node and also weakly connected globally with all other neighbor nodes, enabling nonlin-

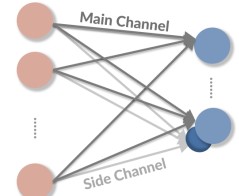

Figure 3: The duo-channel interactions.

ear and high-order information aggregation, which is then forwarded to the output nodes along with the original real-valued information. This enhancement allows for more nuanced representations of the system's energy landscape, expanding the model's capacity to capture intricate patterns and behaviors in complex problems. Particularly, we redefine the node dynamics as:

$$\frac{d\sigma_i}{dt} = \sum_{j=1}^{M} J_{ij}\sigma_j + \sum_{k=1}^{K} N_{ik}\varphi\left(s_k\right) - 2h_i\sigma_i. \tag{5}$$

Here, $\varphi\left(s_k\right) = \varphi\left(\sum_{j=1}^{M} M_{kj}\sigma_j\right)$ represents the polarized shadow nodes used to learn the high-order nonlinear node interactions. K is the number of shadow nodes. $\varphi$ represents a non-linear function introduced to achieve polarization. For a hardware-friendly design, we use an adjustable piecewise function, defined as $\varphi(x) = -1$, if $x < -\theta$; $\varphi(x) = 1$, if $x > \theta$; $\varphi(x) = \tau x$, if $-\theta \leq x \leq \theta$. Our proposed enriched node dynamics significantly enhances the model expressivity from three perspectives:

- *Duo-channel Non-linear Node Interaction:* The shadow nodes, polarized by learnable piecewise functions, create enriched node dynamics. This enables system evolution powered by duo-channel non-linear node interaction. Moreover, the learnable parameter $\tau$ in the piecewise function enables fine-grained tuning of the system's more refined energy landscape.

- *High-order Interactions:* The designed polarized shadow nodes $\varphi\left(s_k\right)$, $k = 1, 2, ..., K$, hierarchically capture high-order interactions among local and remote nodes through a learnable matrix $M_{kj}$. This enables the system to capture intricate, non-local dependencies in the data distribution.

- Adaptive Complexity: The quantity of these shadow nodes can be dynamically adjusted to align with the complexity of the targeted problems, further enhancing the system's adaptability.

***The Physical Embodiment of DS-Solver.*** The physical embodiment of the enhanced dynamical system model closely resembles that of the current SOTA implementation as illustrated in Figure 4 (as $K \ll M$, the hardware augmentation is minimal). The embodiment for the first and third terms in the enhanced node dynamics remains identical to the SOTA model: with node values mapped to capacitor voltages and with parameters **J** and **h** configured as resistor conductances. This configuration realizes both terms as electrical currents: the flow-in current and the internal current of a node unit, respectively. Following the same design strategy, we realize the second term by embodying $\boldsymbol{\theta}$ as capacitors in the shadow element of the node unit, and **N**, **M**, $\boldsymbol{\tau}$ as resistor conductances. Consequently, the second term is also realized as electrical current, introducing an additional flow-in current to the node unit. The summation of the first two terms, $\sum_{j=1}^{M} J_{ij}\sigma_j + \sum_{k=1}^{K} N_{ik}\varphi(s_k)$, corresponds to the electrical current jointly flowing into the node unit associated with $\sigma_i$, denoted as $I_i^{in}$. Equilibrium for an individual node is achieved when this inflow current neutralizes the current flowing through the node's resistor $R_i$, denoted as $I_i^R = 2h_i\sigma_i$, which represents the last term in the enriched node dynamics. At this point, the node dynamics $d\sigma_i/dt$ equal zero, indicating node stabilization. The system reaches global equilibrium when all nodes stabilize simultaneously. As with the SOTA model, this global equilibrium corresponds to the dynamical system's lowest energy state and the highest probability state, which ideally should be trained to represent the desired solutions to the target equation. This equilibrium-seeking process is governed by the spontaneous energy decrease of dynamical systems, as illustrated in equation 3, and is referred to as natural annealing.

### 3.2 INSTANT ON-DEVICE TRAINING POWERED BY INTRINSIC ELECTRIC SIGNAL

***On-Device Instant Training Algorithm Powered by Dynamical System:*** To extend the extraordinary computational power of dynamical systems to the training process, we propose an efficient on-device training method, ***EC-Train***. This novel approach leverages the intrinsic electrical current of the dynamical system as a feedback signal to adjust parameters on-device and further self-reconstruct the energy landscape, precisely mirroring the data distribution of the target problem. EC-Train establishes a well-defined physical entity within the electric dynamical system that functions as a loss mechanism, enabling second-level instant model training directly on the dynamical system processor. This innovation significantly reduces training costs by orders of magnitude compared to conventional offline training methods executed on digital processors.

The development of EC-Train is founded on a key observation: A perfectly trained DS-Solver should achieve equilibrium when its nodes are set to the ground truth values from the training dataset. At equilibrium, the aggregate electric current $I_i^{in} = \sum_{j=1}^{M} J_{ij}\sigma_j + \sum_{k=1}^{K} N_{ik}\varphi(s_k)$ flowing into node $\sigma_i$ must neutralize its internal resistor current $I_i^R = 2h_i\sigma_i$, thereby stabilizing the capacitor voltage that represents the node's value. Consequently, the on-device training process of EC-Train aims to minimize the difference between these currents $(I_i^{in} - I_i^R)$ for all nodes when set to ground truth values. The EC-Train loss function can be formulated as:

$$L = \frac{1}{N}\sum_{i=1}^{N}(I_i^{in} - I_i^R)^2. \tag{6}$$

The electric currents provide feedback signals for each node:

$$\delta_i = \frac{\partial L}{\partial \sigma_i} = -\frac{2}{N}(I_i^{in} - I_i^R). \tag{7}$$

The updates for the trainable parameters are derived as the gradients of the feedback signal with respect to each parameter. Specifically, the trainable coupling parameters include **J**, **N**, and **M**. **h** serves as a set of scaling factors and is fixed as a constant. Therefore, the gradients with respect to $J_{ij}$, $N_{ik}$, and $M_{kj}$ are:

$$\nabla J_{ij} = \delta_i \cdot \sigma_j; \ \nabla N_{ik} = \delta_i \cdot \varphi(s_k); \ \nabla M_{kj} = \tau \sum_i \delta_i N_{ik}\sigma_j. \tag{8}$$

***The Physical Embodiment of DS-Solver with EC-Train*** is illustrated in Figure 4. We introduce a lightweight yet effective modification to enable parameter adjustments based on feedback from electrical currents: an additional feedback signal path (highlighted in brown) for each parameter, connecting the node unit and its shadow element to their corresponding parameters, realized as resistors within coupling units. These feedback paths allow the electronic dynamical system to propagate signals to the coupling units, facilitating instantaneous parameter adjustment through rapid charging or discharging of the programmable resistors. With EC-Train, the system performs infinite updates within each natural annealing cycle, continuously and instantly reshaping the energy landscape to achieve convergence at "speed of electrons", at negligible cost compared to traditional training on digital processors. The EC-Train training process is as:

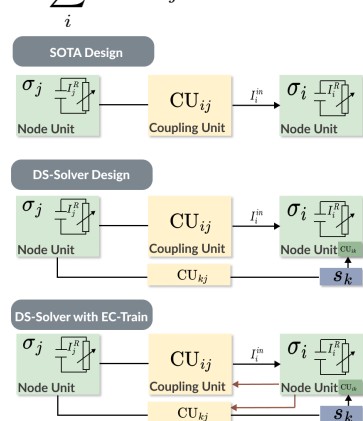

Figure 4: Architecture design of DS-Solver with EC-Train.

1. *Initialization:* The capacitor voltages representing node values are set to their ground truth values, while the trainable parameters are randomly initialized.

2. *Natural Annealing:* The system undergoes a rapid, spontaneous energy decrease, driving it toward equilibrium and generating the electrical current $I_i^{in} - I_i^R$, which serves as the feedback signal to adjust the system parameters.

3. *Parameter Adjustment:* The trainable parameters are updated based on the feedback signal.

4. *Continuous and Iterative Training:* The update of trainable parameters results in a new electrical current $I^{in}$, which flows back to the node units, updating the feedback signal $I_i^{in} - I_i^R$, and instantaneously initiating a new training iteration. This process continues iteratively across the entire training set until convergence is achieved.

## 4 EVALUATION

### 4.1 EXPERIMENT SETUP

As a pioneering effort demonstrating the significant potential of dynamical systems, we evaluate the performance of DS-Solver across various domains, including PDE solving in scientific computing, Transformer approximation in ML, and hard-to-define equation solving in real-world problems.

**Datasets and Baselines.** For PDE solving, we consider PDEs that commonly exist in the physical world, including Heat, Wave, Laplace, Poisson, Navier Stokes, and Schrödinger equations. For each of the PDEs, datasets are generated using the finite difference method, with unique initial conditions, boundary conditions, and domain geometries (more details are provided in the Appendix). We compare the proposed DS-Solver with Multi-Layer Perceptrons (MLP) (Rumelhart et al., 1986), Radial Basis Function Networks (RBF) (Lowe & Broomhead, 1988), Support Vector Machines (SVM) (Cortes, 1995), Kolmogorov–Arnold Networks (KAN) (Liu et al., 2024), and the current SOTA dynamical system based method NP-GL (Wu et al., 2024). Detailed implementation configurations are provided in the Appendix.

For Transformer approximation, we demonstrate DS-Solver's effectiveness in approximating key components of the GPT-2 model (124M parameter version) (Wolf, 2019), specifically the first multi-head self-attention layer and the first decoder block. We extract a subset (∼60,000 tokens) of input-output pairs from the first *self-attention* layer and the first *decoder* block in the pretrained GPT-2 using the OpenWebText training set. We compare DS-Solver with the SOTA dynamical system based method NP-GL (Wu et al., 2024). DS-Solver and NP-GL are trained on the constructed datasets to replicate the complex, non-linear transformations between inputs and outputs of the selected components. Detailed implementations are in the Appendix.

For hard-to-define equation solving in real-world problems, we evaluate the performance of DS-Solver in spatial-temporal prediction tasks and the electric field energy prediction task in nuclear fusion. Regarding spatial-temporal prediction, we evaluate the proposed DS-Solver on six real-world datasets for four applications. (1) Traffic flow prediction with two datasets PEMS04 and PEM08 (Chen et al., 2001). (2) Air quality prediction including PM2.5 and PM10 (Kong et al., 2021). (3) Taxi demand prediction (NYC Taxi): predicting the hourly number of taxi trips (New York City Taxi and Limousine Commission, 2024). (4) Pandemic progression prediction (Texas COVID): predicting the daily number of new cases (Centers for Disease

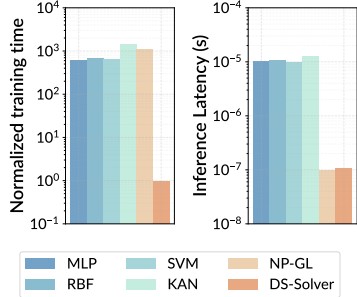

Figure 5: Training time and inference latency for PDE solving.

Control and Prevention, 2024). We compare DS-Solver with SOTA spatial-temporal prediction baselines, including Graph WaveNet (Wu et al., 2019), MTGNN (Wu et al., 2020), DDGCRN (Weng et al., 2023), MegaCRN (Jiang et al., 2023), and the SOTA dynamical system based method NP-GL (Wu et al., 2024). Detailed implementations are in the Appendix.

Electric field energy prediction is a task from nuclear fusion research, which helps to optimize fusion reactions occurring in extremely complex physical dynamical systems. To evaluate DS-Solver on this task, we construct a dataset using Particle-in-Cell (PIC) simulations (Fonseca et al., 2002). Our setup encompasses key fusion-relevant parameters, with the prediction task formulated as follows: given the input features Electron Temperature (Te), Ion Temperature (Ti), Laser Intensity (Li), and time $t$, forecast the corresponding electric field energy (This dataset will be open-sourced to facilitate further research in this critical domain). We compare DS-Solver with the SOTA dynamical system based method NP-GL (Wu et al., 2024). Detailed implementations are in the Appendix.

**Experimental Platforms.** We conduct our experiments using an NVIDIA A100 40GB SXM GPU for non-dynamical system based baselines, measuring total training time, inference latency per sample, accuracy, and energy consumption. For the SOTA dynamical system based baseline NP-GL, we use the same A100 GPU for training time measurement, while employing its proposed dynamical system for inference latency and accuracy evaluation. The proposed DS-Solver is assessed using a custom CUDA-based Finite Element Analysis (FEA) software simulator, built upon the BRIM framework (Afoakwa et al., 2021), for training time, inference latency, and accuracy measurements.

Table 1: PDE solution comparison in MAE, best results are in bold.

| Dataset | Heat | Wave | Laplace | Poisson | Navier Stokes | Schrödinger |
|---|---|---|---|---|---|---|
| MLP | 3.7e-4 | 6.5e-4 | 4.8e-4 | 1.3e-4 | 7.1e-4 | 4.2e-4 |
| RBF | 5.3e-4 | 6.2e-4 | 3.9e-4 | 1.0e-4 | 5.8e-4 | 3.5e-4 |
| SVM | 6.5e-4 | 7.1e-4 | 5.9e-4 | 8.2e-5 | 5.3e-4 | 4.1e-4 |
| KAN | 2.1e-5 | 2.7e-5 | 1.9e-5 | 7.2e-6 | 3.1e-5 | 4.3e-5 |
| NP-GL | 2.8e-4 | 4.9e-4 | 3.7e-5 | 8.3e-5 | 1.6e-4 | 4.8e-4 |
| DS-Solver | **1.8e-5** | **2.2e-5** | **1.6e-5** | **6.4e-6** | **2.7e-5** | **2.5e-5** |

Table 2: Attention layer replacement

| Dataset | LAMBADA | WT2 | WT103 |
|---|---|---|---|
| GPT-2 | 35.13 | 29.41 | 37.50 |
| NP-GL | 41.56 | 34.82 | 45.27 |
| DS-Solver | 35.38 | 29.64 | 37.82 |

Table 3: Decoder block replacement

| Dataset | LAMBADA | WT2 | WT103 |
|---|---|---|---|
| GPT-2 | 35.13 | 29.41 | 37.50 |
| NP-GL | 42.95 | 35.72 | 46.39 |
| DS-Solver | 36.46 | 30.15 | 38.02 |

DS-Solver's energy consumption was evaluated using the Cadence Mixed-Signal Design Environment with 45nm CMOS technology.

## 4.2 EXPERIMENTAL RESULTS

**PDE solving.** We compare DS-Solver and baselines on the selected PDEs. The best MAE achieved by each method is presented in Table 1. The results demonstrate that DS-Solver consistently outperforms traditional methods across all PDEs. Besides, we average each model's training time and inference latency across all PDEs, the comparison is shown in Figure 5. DS-Solver shows extraordinary training and inference efficiency compared to traditional methods implemented on digital processors, achieving $897\times$ training speedup and over $101\times$ inference speedup on average.

**Transformer Approximation.** We conduct two separate evaluations to evaluate the performance of integrating the trained NP-GL and DS-Solver models into GPT-2. In the *first* evaluation, we replace the first self-attention layer in GPT-2 with either NP-GL or DS-Solver and measure the resulting systems' performance on the LAMBADA (Paperno et al., 2016), WikiText2 (WT2), and WikiText103 (WT103) (Merity et al., 2016) datasets using the perplexity (PPL) metric. In the *second* evaluation, we substitute the first decoder block with NP-GL or DS-Solver and similarly evaluated performance across the same datasets. As shown in Tables 2 and 3, replacing the first self-attention layer with the trained NP-GL resulted in a substantial increase in PPL, with an average increase of 6.54 across these datasets. Similarly, substituting the first decoder block with NP-GL leads to an average PPL increase of 7.67. In contrast, DS-Solver maintained PPL scores much closer to the original GPT-2, with only a small average increase of 0.27 when replacing the first self-attention layer and 0.86 when replacing the first decoder block across all datasets. These results demonstrate the superior capability of DS-Solver compared to NP-GL in learning complex transformations within ML models. As shown in Figure 6, DS-Solver achieves an average speedup of 73.2× on the self-attention and decoder layers compared to the baselines on GPU.

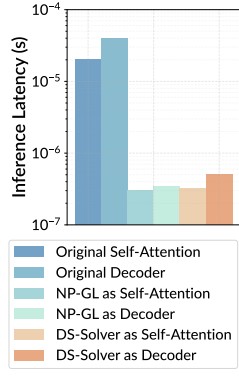

Figure 6: Inference latency for Transformer.

**Spatial-Temporal Prediction.** We present the test MAE of baselines and DS-Solver on selected datasets in Table 4, where lower values indicate better performance. The results show that DS-Solver outperforms all baselines across all datasets. Figure 7 shows the training time and inference latency comparisons. DS-Solver demonstrates substantial computational efficiency, consistently delivering orders of magnitude speedup in training time across all datasets. The training speedup ranges from $478\times$ to $2408\times$ compared to the best baseline NP-GL, while delivering an average of $886\times$ training speedup versus all baselines and $1923\times$ inference speedup versus the baselines executed on GPU.

**Electric Field Energy Prediction.** We compare DS-Solver with the current SOTA dynamical system based method NP-GL. The test MAE and RMSE are presented in Table 5, showing the performance of NP-GL and DS-Solver in predicting electric field energy along two orthogonal directions (E1 and E2).

Table 5: Electric field energy prediction.

| Dataset | E1 | | E2 | |
|---|---|---|---|---|
| | MAE | RMSE | MAE | RMSE |
| NP-GL | 3.75e-2 | 4.28e-2 | 5.31e-2 | 5.84e-2 |
| DS-Solver | 1.13e-2 | 1.64e-2 | 3.17e-2 | 3.92e-2 |

DS-Solver achieved impressively low test MAE and

Table 4: Spatial-temporal prediction comparison in MAE, best results are in bold.

| Dataset | PEMS04 | PEMS08 | PM2.5 | PM10 | NYC Taxi | Texas Covid |
|---------|--------|--------|-------|------|----------|-------------|
| Graph WaveNet | 20.84 | 15.77 | 1.823 | 1.954 | 10.22 | 82.96 |
| MTGNN | 19.96 | 15.15 | 1.833 | 1.990 | 7.079 | 84.17 |
| DDGCRN | 18.97 | 14.64 | 1.711 | 1.881 | 3.059 | 23.94 |
| MegaCRN | 17.65 | 13.70 | 1.646 | 1.741 | 6.082 | 83.73 |
| NP-GL | 17.07 | 13.51 | 1.624 | 1.730 | 3.031 | 22.04 |
| DS-Solver | **16.97** | **13.50** | **1.565** | **1.653** | **2.488** | **17.31** |

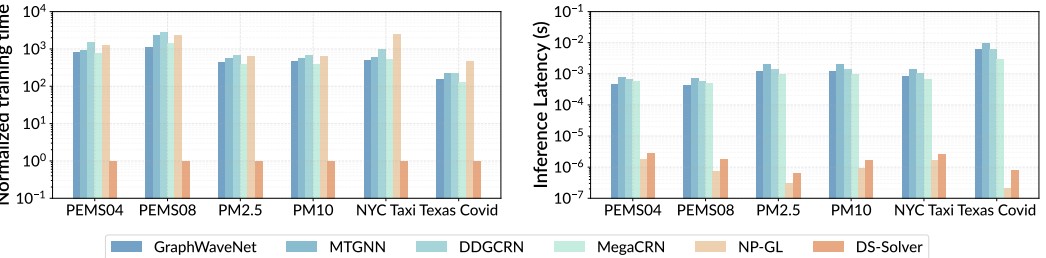

Figure 7: Training time and inference latency comparison for spatial-temporal prediction.

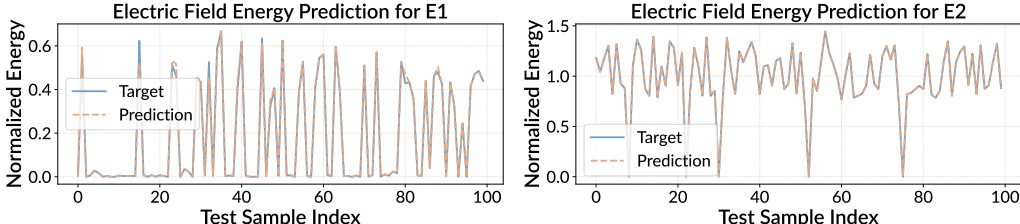

Figure 8: Electric field energy prediction for E1 and E2.

RMSE with the dataset normalized to [0,1], outperforming NP-GL on both E1 and E2. These minimal MAE values underscore DS-Solver's effectiveness in capturing and reproducing the intricate dynamics of electric field energy evolution in fusion simulations, highlighting its potential as a powerful tool for plasma physics research and fusion reactor design optimization. Besides, the visualizations of DS-Solver's predictions and ground truths are provided in Figure 8, elucidating DS-Solver's remarkable ability to accurately forecast electric field energy across a diverse range of simulation timescales and physical parameters.

**Power and Energy Efficiency.** DS-Solver provides ultra-low power of 1.6W for training, and 326mW for inference. For a reasonable reference, we assume the average power for the GPU used in this work is 250W. In terms of overall energy consumption, taking into account the exceptional speedups achieved in training and inference, DS-Solver achieves approximately, on average, $1.40 \times 10^5$ and $1.38 \times 10^5$ higher energy efficiency in training for PDE solving and spatial-temporal prediction applications, respectively; $7.74 \times 10^4$, $5.61 \times 10^4$, $1.47 \times 10^6$ higher energy efficiency in inference for PDE solving, Transformer approximation, and spatial-temporal prediction, respectively.

## 5 CONCLUSION

While modern machine learning methods excel as equation solvers, their growing complexity and substantial operational costs pose challenges to sustainable development. In contrast, nature effortlessly solves complex equations through dynamical systems that naturally evolve towards low-energy states without explicit guidance. In response, we introduce DS-Solver, a nature-powered AI paradigm that leverages a self-trainable, expressive dynamical system capable of solving a wide range of equations with remarkable efficiency. Experimental results across key equations from various domains show that DS-Solver achieves 42% higher accuracy than the current SOTA – while delivering a $\sim 10^3 \times$ speedup and $\sim 10^5 \times$ energy efficiency compared to traditional neural network solutions on GPUs for both inference and training. These results highlight its broad impact on overcoming the persistent computational bottlenecks across various critical fields, including ML, scientific exploration, and real-world complex systems.

## REPRODUCIBILITY

We have a comprehensive plan to enable reproducibility. (a) The dynamical system processor has been manufactured, and we have access to the real hardware. We plan to provide the public with remote access to the processor through the university's computing cluster. (b) We have developed an accurate GPU-based emulator of the dynamical system processor. This software will be open sourced, enabling the reproducibility of this work and open research even without physical access to the hardware. The integration of the hardware in our paper with existing systems (e.g. GPUs) can be seamless since it is fully CMOS-based, utilizing the same underlying chip technology as GPUs, CPUs, and FPGAs.

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

# A  APPENDIX

## A.1  DATASETS

For the PDE solving tasks, we generated datasets for the following equations: Heat equation, Wave equation, Laplace equation, Poisson equation, Navier-Stokes equation, and Schrodinger equation. For each PDE, we used the finite difference method to discretize the equations over specific domain geometry, initial conditions, and boundary conditions. We generate one-dimensional heat equation data with spatial domain $x \in [0, 1]$, temporal domain $t \in [0, 1]$, initial condition $u(x, 0) = \sin(\pi x)$, and Dirichlet boundary conditions. We generate one-dimensional wave equation data with spatial domain: $x \in [0, 1]$, temporal domain $t \in [0, 1]$, initial conditions $u(x, 0) = \sin(\pi x)$, $\frac{\partial u}{\partial t}|_{t=0} = 0$, and Dirichlet boundary conditions. We generate the two-dimensional Laplace equation data with spatial domain $x \in [0, 1]$, $y \in [0, 1]$, boundary conditions: $u(0, y) = 1$, $u(L, y) = 0$, $u(x, 0) = 0$, $u(x, L) = 0$. We generate the two-dimensional Poisson equation data with spatial domain $x \in [0, 1]$, $y \in [0, 1]$, and Dirichlet boundary conditions. We generate the two-dimensional Navier-Stokes equation data with spatial domain $x \in [0, 1]$, $y \in [0, 1]$, initial velocity $u = 0$, $v = 0$, initial pressure $p = 0$, no-slip conditions on the walls, bottom and side walls: $u = 0$, $v = 0$, and top lid: $u = 1$, $v = 0$. We generate the one-dimensional time-dependent Schrödinger equation data with spatial domain: $x \in \left[-\frac{1}{2}, \frac{1}{2}\right]$, and periodic boundary conditions: $\psi\left(-\frac{1}{2}, t\right) = \psi\left(\frac{1}{2}, t\right)$. All simulations were run until the convergence criteria were met.

For PDE solving, and electric field energy prediction tasks, we split the generated data into 60% training, 20% validation, 20% testing. For the spatial-temporal prediction task, we follow the settings in (Wu et al., 2024) to split the data into 70% training, 20% validation, 10% testing.

## A.2  MODEL IMPLEMENTATION DETAILS

For PDE solving, we use the same settings as in (Wu et al., 2024) for NP-GL, while other baselines employed ReLU activation, Adam optimizer, and two-layer architectures. Hyperparameter searches were conducted for MLP (batch size: [32,64,128], hidden neurons: [8,16,32]), RBF with Gaussian basis functions (batch size: [32,64,128], centers: [8,16,32]), SVM with polynomial kernel (batch size: [32,64,128]), and KAN (spline order 3, grids: [5,10,15,20], hidden neurons: [8,16,32]). DS-Solver was simulated with 4 and 8 shadow nodes. In spatial-temporal prediction, all baselines adhered to settings from their original papers, with DS-Solver utilizing 32 shadow nodes. For electric field energy prediction, we use the same settings as in (Wu et al., 2024) for NP-GL, and implemented 4 shadow nodes for DS-Solver.

