# OpenReview forum: "A Brain-Inspired Machine Learning Paradigm for Nature-Powered Equation Solving"
_ICLR.cc/2025/Conference — ICLR 2025 Conference Withdrawn Submission_

### Official Review · Reviewer_q5z6 · 2024-10-27

**Soundness:** 1
**Presentation:** 1
**Contribution:** 1
**Rating:** 1
**Confidence:** 4

**Summary:**

This study emphasizes "brain-inspired" or "nature-powered" design and claims to achieve computational efficiency surpassing SOTA through these attributes, yet it essentially proposes a kind of improved algorithm for an analog circuit implementation of a continuous Hopfield network (Ising model).

**Strengths:**

With the recent Nobel Prize in Physics, renewed attention to Hopfield networks is indeed timely and could provide fresh perspectives.

**Weaknesses:**

Describing Hopfield networks as a SOTA method is questionable--particularly as they have limited applicability in modern contexts. Furthermore, the conceptual parallels between ising model and physical or brain-like systems have been debated for nearly half a century, but any new insights were not introduced here.

The study presents PDE-solving as a means to evaluate performance, but the task specifics are unclear. It remains ambiguous whether the model is learning PDEs from data (as in a Neural PDE Solver) or using NNs as bases of PDE solutions (akin to PINNs). Overall, the intent to improve upon NP-GL (Wu et al., 2024) is evident, but it did not evaluate PDEs and focused on GNN-related tasks. Thus, describing NP-GL as SOTA is inappropriate. Other comparison methods include outdated techniques like RBFs and SVMs, bypassing contemporary machine learning methods and thus casting doubt on reliability. While comparisons in Table 3 seem more reasonable, the rationale for selecting these methods is unclear.

In Table 2, there is an evaluation of partially replacing Transformer components, but only power consumption is reported, with no performance metrics. This inconsistency raises concerns that only favorable aspects from various tasks are cherry-picked for presentation.

There are also questions regarding the feasibility of electrical circuit implementation. Regardless of whether one uses Transformers or other deep learning methods, their structure is optimized for parallel computation on GPUs, making them suboptimal for custom circuit design. Even with circuit miniaturization and implementation efficiency, can the proposed architecture truly compete with GPUs? Additionally, when combining this architecture with Transformers, how would the parts that the architecture cannot replace be handled?

**Questions:**

See weaknesses.

---

### Official Review · Reviewer_GfiJ · 2024-10-27

**Soundness:** 1
**Presentation:** 2
**Contribution:** 1
**Rating:** 3
**Confidence:** 4

**Summary:**

Authors consider a trainable physical system based on an Ising model which can be trained much faster and in a much more energy efficient manner than digital systems. They also present benchmarks showing orders of magnitude improvements in latency and energy efficiency for inference on various tasks, including PDE solving and language modeling.

**Strengths:**

One strength of the paper is that they motivate well the problem they are trying to solve, namely, how to design a physical system that is both expressive and energy-efficient, and which does not involve complicated procedures for training (where a digital model of the system is used to train, for example, which often leads to degraded performance).

**Weaknesses:**

In order of importance, the paper has the following weaknesses:
- The implementation of the DS-solver is unclear. It is based on analog hardware, and therefore there should be much more details on what the components actually are and what realistic effects were/weren't taken into account in the experiments (see Questions).
- One of the main claims of the paper is that the designed Ising-like physical system allows for high expressivity. Why is this the case exactly? For example, how can we be sure that it can be as expressive as convolutions or attention? Further theoretical or empirical investigation would be needed to make this claim stronger.
- For the transformer benchmarks, if only the first self-attention layer is accelerated, surely this results in an overall latency improvement that is quite marginal (due to Amdahl's law). This makes the result much less impressive, and it would be good to add and end-to-end evaluation of latency for the full model (including I/O times and energy consumption under realistic assumptions). Even then, I am not convinced that you can replace a self-attention layer with your physical system and maintain the same outputs with high precision in a realistic implementation (at least it was not shown precisely how in the paper, and how this depends on device and model parameters).
- The main text refers to implementation details being present in the appendix, but there are very few details there (and none about the language modeling experiments). The fact that the code is not available here is also a problem because of the lack of technical detail throughout the paper.
- What happens in the presence of realistic effects in physical hardware, such as limited resolution, dynamic range, various noise sources and device mismatch. This should be quite central in the discussion, and it is not explained what you considered in your simulations.
- Many results and claims are incomplete. For example, it is written "DS-Solver provides ultra-low power of 1.6W for training, and
326mW for inference". For what task and system size, and what device specifications?
- There is a whole body of work around [equilibrium propagation](https://www.frontiersin.org/journals/computational-neuroscience/articles/10.3389/fncom.2017.00024/full), which was not referred to in the related work or throughout the paper, which is very similar to what is achieved here, with stronger theoretical justification and experimental results. A discussion of the benefits of your approach with respect to equilibrium propagation would be interesting to add.

**Questions:**

You mention: ``The proposed DS-Solver is assessed using a custom CUDA-based Finite Element Analysis (FEA) software simulator, built upon the BRIM framework (Afoakwa et al., 2021), for training time, inference latency, and accuracy measurements." In another section, you mention "DS-Solver’s energy consumption was evaluated using the Cadence Mixed-Signal Design Environment with 45nm CMOS technology."
What did you use to simulate your system then? And why is the FEA useful? Having a simple simple custom differential equation solver based on Jax or Pytorch would have been helpful to understand/useful to scale to large system sizes.

---

### Official Review · Reviewer_YPUs · 2024-10-30

**Soundness:** 3
**Presentation:** 3
**Contribution:** 3
**Rating:** 6
**Confidence:** 3

**Summary:**

This paper presents DS-Solver, a nature-powered AI system that employs highly expressive and self-trainable dynamical systems to efficiently and accurately solve various types of equations. The paper makes two main contributions: (1) Enhancing system expressivity by introducing coupled real-valued and polarized shadow nodes, and (2) Proposing an on-device learning method that uses intrinsic electrical signals as loss function, enabling instant self-training of the dynamical system at minimal cost. Results show that DS-Solver achieves 42% higher accuracy than SOTA while offering orders-of-magnitude improvements in both inference and training speed compared to traditional neural network solutions on GPUs.

**Strengths:**

1. The proposed DS-Solver achieves 42% higher accuracy than current SOTA, with approximately 1000x speedup and 100,000x improvement in energy efficiency.
2. This paper can solve different types of equations across multiple domains, including partial differential equations in scientific computing.
3. This paper introduces a new AI paradigm based on principles found in nature.

**Weaknesses:**

1. The paper lacks discussion on the impact of shadow node count on performance and guidelines for determining the optimal number of shadow nodes
2. The feedback signal path implementation shown in Figure 4 is oversimplified and requires more detailed explanation.
3. The visualization in Figure 8 lacks a detailed error analysis, especially in the small error range, and it is difficult to see the fit quality.

**Questions:**

1.The paper mentions different numbers of shadow nodes being used across experiments, but lacks justification for these choices. Could you explain:
     1) How did you determine these specific numbers?
      2) What is the relationship between problem complexity and optimal shadow node count?
2. For the EC-Train approach on the device, although this paper demonstrates its efficiency advantages, there are still some problems:
How sensitive is the training process to electrical noise in actual hardware implementations? Have you conducted long-term stability tests?

---

### Official Review · Reviewer_6t6U · 2024-11-04

**Soundness:** 3
**Presentation:** 2
**Contribution:** 2
**Rating:** 6
**Confidence:** 2

**Summary:**

This paper proposes the DS-Solver approach, for solving equations using a nature-powered AI paradigm. The method leverages dynamical systems inspired by nature. The experimental results across multiple domains are comprehensive and show promising performance improvements over existing state-of-the-art methods.

**Strengths:**

1.The proposed DS-Solver paradigm is innovative in combining an expressive dynamical system with on-device self-training. The concept of enriching node dynamics with coupled real-valued and polarized shadow nodes enhances the system's expressivity and ability to capture complex interactions.

2.The authors conduct many experiments across the PDE solving, transformer approximation,spatial-temporal prediction, electric field energy prediction,training time and inference latency comparison, allows for a thorough understanding of the method's advantages.

**Weaknesses:**

1.Clarity of the Methodology:While the overall methodology is described, some aspects could be presented more clearly. For example, the physical embodiment of the DS-Solver with EC-Train could be explained in more detail. The equations and derivations related to the node dynamics and training algorithm could be accompanied by more intuitive explanations to aid understanding.

2. There is a lack of theoretical analysis. The article claims to enhance the expressiveness of existing dynamic system AI paradigms, however, there is no theoretical analysis provided to support this claim.

**Questions:**

Could you add a comparison and discussion with the large language model (LLM), which is also capable of performing equation solving? How does this method compare to LLM in terms of accuracy and other performance aspects?

---

### Note · Authors · 2024-11-15

I have read and agree with the venue's withdrawal policy on behalf of myself and my co-authors.